# Experimental Infection of Mid-Gestation Pregnant Female and Intact Male Sheep with Zika Virus

**DOI:** 10.3390/v12030291

**Published:** 2020-03-07

**Authors:** Erika R. Schwarz, Lilian J. Oliveira, Francesco Bonfante, Ruiyu Pu, Malgorzata A. Pozor, N. James Maclachlan, Sarah Beachboard, Kelli L. Barr, Maureen T. Long

**Affiliations:** 1Department of Comparative, Diagnostic, and Population Medicine, College of Veterinary Medicine, University of Florida, Gainesville, FL 32608, USA; eschwarz@ufl.edu (E.R.S.); pur@ufl.edu (R.P.); sebeach@ufl.edu (S.B.); 2Department of Infectious Diseases and Immunology, College of Veterinary Medicine, University of Florida, Gainesville, FL 32608, USA; lijoli@ufl.edu; 3Laboratory of Experimental Animal Models, Division of Comparative Biomedical Sciences, Instituto Zooprofilattico Sperimentale delle Venezie, 35020 Legnaro, Italy; fbonfante@izsvenezie.it; 4Department of Large Animal Clinical Sciences, College of Veterinary Medicine, University of Florida, Gainesville, FL 32611, USA; pozorm@ufl.edu; 5Department of Pathology, Microbiology, and Immunology, School of Veterinary Medicine, University of California, Davis, CA 95616, USA; njmaclachlan@ucdavis.edu; 6Department of Biology, College of Arts and Sciences, Baylor University, Waco, TX 76798, USA; kelli_barr@baylor.edu

**Keywords:** Zika virus, ovine, mid-gestation, pregnancy, testes, animal mode

## Abstract

Zika virus (ZIKV) is an arbovirus that causes birth defects, persistent male infection, and sexual transmission in humans. The purpose of this study was to continue the development of an ovine ZIKV infection model; thus, two experiments were undertaken. In the first experiment, we built on previous pregnant sheep experiments by developing a mid-gestation model of ZIKV infection. Four pregnant sheep were challenged with ZIKV at 57–64 days gestation; two animals served as controls. After 13–15 days (corresponding with 70–79 days of gestation), one control and two infected animals were euthanized; the remaining animals were euthanized at 20–22 days post-infection (corresponding with 77–86 days of gestation). In the second experiment, six sexually mature, intact, male sheep were challenged with ZIKV and two animals served as controls. Infected animals were serially euthanized on days 2–6 and day 9 post-infection with the goal of isolating ZIKV from the male reproductive tract. In the mid-gestation study, virus was detected in maternal placenta and spleen, and in fetal organs, including the brains, spleens/liver, and umbilicus of infected fetuses. Fetuses from infected animals had visibly misshapen heads and morphometrics revealed significantly smaller head sizes in infected fetuses when compared to controls. Placental pathology was evident in infected dams. In the male experiment, ZIKV was detected in the spleen, liver, testes/epididymides, and accessory sex glands of infected animals. Results from both experiments indicate that mid-gestation ewes can be infected with ZIKV with subsequent disruption of fetal development and that intact male sheep are susceptible to ZIKV infection and viral dissemination and replication occurs in highly vascular tissues (including those of the male reproductive tract).

## 1. Introduction

Zika virus (ZIKV) is one of the only arthropod-borne flaviviruses known to be transmitted sexually amongst humans [1]. ZIKV is most notorious for resulting in a variety of birth defects and negative gestational outcomes when infection occurs during pregnancy. In humans, ZIKV is believed to result in trimester-specific effects to the fetus [2,3,4,5,6,7]. Established animal models of ZIKV infection during pregnancy that result in vertical transmission include non-human primates (NHP) and type-1 interferon/interferon receptor deficient mice. In NHP models, first trimester infection most frequently results in fetal demise and reduced fetal development even in asymptomatic mothers, while second trimester infections tend to produce fetuses with higher viral loads but greater fetal viability [8,9,10]. In all NHP models, fetal pathology ranges from mild to severe manifestations, consistent with Congenital Zika Syndrome (CZS) seen in humans [11,12,13,14,15,16]. While NHP models of ZIKV infection during pregnancy recapitulate human infection, other outbred host models can serve as surrogates when ethical and economic constraints create obstacles to the use of NHP.

Sheep offer a unique animal model for translational biomedical research, and have long been used as a model for human pregnancy and fetal development due to longer gestation and comparably staged rates of fetal development [17,18]. The gestational length of sheep is approximately half of that of humans (~150–155 days), with fetal growth rates paralleling those seen in human fetuses for the first two trimesters of pregnancy. Sheep are also highly susceptible to many teratogenic viruses; mid-gestation infection of sheep with viruses similar to ZIKV (such as bluetongue, Wesselbron, Cache Valley fever, Aino, Akabane, and Schmallenberg) typically results in pathology ranging from severe congenital defects and abortion to chronic viral carriers at birth [19,20,21,22,23,24]. The feasibility of sheep as a model for ZIKV infection has been established. We have shown that early gestation ZIKV infection of pregnant sheep results in detectable virus and a robust immune response by the ewes, with femur-sparing patterns of fetal growth retardation and changes in gestational outcomes in the absence of maternal clinical signs of infection [25]. The effects of ZIKV infection seen during the first trimester in sheep are comparable to those observed in humans, as well as fetal outcomes caused by ovine infections; primarily fetal loss [19,20,21,22,23,24]. Furthermore, previous research indicates that the viral kinetics of ZIKV in fetal sheep cells has been shown to be similar to that in immortalized *Aedes albopictus* cells [26]. We have also demonstrated that immortalized adult sheep kidney cells and immortalized fetal sheep testicular cells are susceptible to ZIKV infection and can sustain viral replication for many days [25].

ZIKV is also unique in that sexual transmission occurs with infection of the male reproductive tract with possible establishment of prolonged, asymptomatic infection in men [27,28,29,30]. This carrier status in males may make men more likely to sexually transmit ZIKV to females than vice versa [1]. Whether or not male ZIKV infection negatively affects male fertility has yet to be answered [31,32]. Only immunodeficient mice have been developed as models of male ZIKV infection; however, because of the confounding immune status of these animals, this model does not recapitulate infection in men. These animals become infected with ZIKV and high viral loads can be isolated from many organs including the testes [29,33,34,35,36]. 

Male sheep and humans share similar reproductive anatomy, including the presence of comparable accessory sex glands [37]. Thus, rams have been used as a model for the human male reproductive system, particularly with regards to endocrinology and embryology [38,39,40,41]. Recent work has shown that experimental infection of intact male sheep with bluetongue virus, a natural arbovirus of sheep, resulted in localization of the virus in the male reproductive tract and subsequent testicular degeneration, similar to pathology that can occur in ZIKV infection of male humans [42,43]. Similarly, virus has been isolated from the reproductive tract of rams infected with border disease virus (BDV) [44,45]. Given that viruses appear to readily localize to the male reproductive tract, this species may offer an alternative to immunodeficient animal models in order to study male ZIKV infection. 

Animal models of ZIKV sexual transmission are currently lacking. Sexual transmission from male to female has been demonstrated using interferon receptor deficient mice [46]. While, murine models offer researchers a widely available platform in which to study the effects of male ZIKV infection, NHPs represent a more translatable model. Infection of mature male baboons and macaques leads to localization of ZIKV in reproductive tissue [14,47], while shedding in the semen is additionally apparent in macaques [13,48,49]. However, neither murine nor NHP models offer both wide accessibility and broad translatability as models of male ZIKV infection.

This current work furthers the development of an ovine model of ZIKV infection, examining mid-gestation infection in female sheep and infection in male sheep. Here, we investigated the hypothesis that infection of pregnant sheep at mid-gestation would result in vertical transmission of ZIKV. We also investigated the hypothesis that intact male sheep are susceptible to ZIKV infection and that localization of the virus will occur in the reproductive tract.

## 2. Materials and Methods 

### 2.1. Animals

All animal work was performed under the approval and guidance of the University of Florida Institutional Animal Care and Use Committee (Approval #201609345, October 14, 2106). Eight intact, male, specific pathogen-free (SPF) sheep and six pregnant, female, SPF sheep were purchased (*Ovis aries*, Polypay breed; New England Ovis LLC, Rollinsford, NH, USA). Animals were housed in extended Animal Biosafety Level 2 facilities; control animals were housed separately from infected animals and males were housed separately from females. Plaque reduction neutralization tests (PRNT) revealed no prior exposure to ZIKV or West Nile virus (WNV). Standard practices for feeding, husbandry, and preventive medicine were performed under the guidance of University of Florida Animal Care Services. Baseline physical parameters were recorded daily and all animals were monitored for signs of physical illness and discomfort. All female sheep were confirmed pregnant via transabdominal ultrasound at the time of receipt, and all were between 57 and 64 days of gestation (DG) (corresponding with early second trimester) at the time of viral challenge. All male sheep were between three and four months of age (early sexual maturity) at the time of viral challenge.

### 2.2. Virus Propagation

The Asian lineage ZIKV strain R103451 (NR-50355, BEI Resources, NIAID, NIH, Manassas, VA, USA) was passaged once and expanded upon acquisition in Vero-76 cells (CRL-1587, ATCC, Manassas, VA, USA) and stored in sheep serum at −80 °C. Virus was quantitated by plaque forming unit (PFU) assay, in which ZIKV mixed as eight 10-fold serial dilutions in commercial cell culture media (Modified Eagle’s Media, MEM, Gibco-Thermo Fisher Scientific, Gaithersburg, MD, USA), inoculated into duplicate wells of a 12-well tissue culture plate containing 95% confluent Vero-76 cells and incubated at 37 °C with 5% CO_2_ for 1 h. Inoculum was removed and 1 mL of 0.05% methylcellulose overlay was added to each well. After incubation for 72 h, wells were rinsed and stained with 0.1% Coomassie blue (Thermo Fisher Scientific, Waltham, MA, USA) in 50% methanol, 43% ethanol, and 7% acetic acid for visualization. Plaques were counted and the PFU was calculated as previously described [50]. 

### 2.3. Experimental Design

#### 2.3.1. Experiment #1: Mid-Gestation Pregnant Ewe Infection

Four mid-gestation pregnant sheep were inoculated with two 1 mL injections of 1 × 10^8^ PFU ZIKV intravenously and subcutaneously on day (D) 0 post-infection (PI); two uninfected animals served as controls (Figure 1). A back titration was performed on aliquots of ZIKV inoculum transported in the same way as the infecting inoculum to confirm the dose and stability of the virus. Whole blood and serum were collected daily for the first week PI, then every three days thereafter from all ewes. Transabdominal ultrasonography was performed weekly to monitor the number and viability of the fetuses. At D13 and D20 PI, the two control animals were euthanized; one infected animal was euthanized each day on D14, D15, D21, and D22 PI, and comprehensive post-mortem examinations were immediately performed on all ewes and fetuses.

#### 2.3.2. Experiment #2: Intact Male Infection

Six sexually mature, intact male sheep were inoculated with two 1 mL injections of 1 × 10^7^ PFU ZIKV intravenously and subcutaneously on D0 PI; two uninfected animals served as controls (Figure 2). A back titration was performed on aliquots of ZIKV inoculum transported in the same way as the infecting inoculum to confirm the dose and stability of the virus. One ram was euthanized on D2, D3, D4, D5, D6, and D9 PI and comprehensive post-mortem examinations were immediately performed. Control rams were euthanized on D12 and D14 PI. Serum and whole blood (4 mL in 14.2% Acid Citrate Dextrose [ACD]; 2 mL in EDTA-treated tubes) was collected from the two control rams and the infected animals euthanized on D6 and D9 PI twice daily for the first three DPI, then daily for three more days. Serum and blood were not collected from the animals euthanized on D2, D3, D4, and D5 PI.

### 2.4. Ante-Mortem Sample Processing and Post-Mortem Evaluation and Collection of Tissue

In both experiments, peripheral blood mononuclear cell (PBMC) samples were obtained from whole blood collected in ACD- or EDTA-treated tubes. PBMC were isolated by lysing red blood cells (ACK Lysing Buffer, Gibco), washing (Hanks’ Balanced Salt Solution, HBSS, Gibco), and pelleting by centrifugation at 300× *g* for 5 min at room temperature (RT), then re-suspended with HBSS and separated into aliquots of 1 × 10^6^ cells, which were immediately cultured or used for RNA extraction. Additional aliquots of 1 × 10^6^ cells were cryopreserved at −80 °C in fetal bovine serum (FBS) containing 10% DMSO. During PBMC isolation, plasma was isolated from the whole blood samples and aliquots of serum were stored at −80 °C for future use. 

All animals were humanely euthanized with an overdose of sodium pentobarbital and phenytoin (Beuthanasia-D, Merck Animal Health, Madison, NJ, USA) and necropsies were immediately performed. Placentomes were counted and characterized for all ewes. Fetal morphometric data, including body weight, brain weight, biparietal diameter, nose–occipital length (NOL), crown–rump length (CRL), cranial circumference (CC), and femur length was recorded for all fetuses. From all animals, portions of tissues collected during necropsy were sterilely dissected, sectioned, and snap frozen in liquid nitrogen, then stored at −80 °C until use.

Additional portions of all tissues collected during necropsy were immediately homogenized (Stomacher 80 Biomaster, Seward, West Sussex, United Kingdom). Briefly, individual sections of organs were transferred to sterile stomacher bags with ice-cold serum-free media (MEM, Gibco) and immediately processed on the highest setting until homogenized, and then transferred to sterile 15 mL conical tubes and centrifuged at 1000× *g* for five minutes at 4 °C. From each fetus, pooled tissue samples included (1) the spleen and thymus, and (2) the lung, liver, and kidney. From each male animal, pooled tissue samples included (1) the accessory sex glands (ampullae of the ductus deferentes, vesicular glands, disseminate part of the prostate, and bulbourethral glands), and (2) the testes and epididymides. All other tissues were homogenized separately. Homogenate supernatant was immediately used for viral culture or RNA extraction and the remainder of the homogenized tissue was stored at −80 °C for future use. 

Sections of all collected tissues were preserved in 10% buffered formalin and embedded in paraffin. Slides were stained with hematoxylin and eosin (H&E) and histologic evaluation was performed by a veterinary pathologist. Pregnant ewe tissues analyzed included placentome, intercotyledonary chorioallantois, umbilical cord, liver, spleen, kidney, mammary and adrenal glands, lymph node (i.e., inguinal lymph node), urinary bladder and ovary. Fetal tissues analyzed included liver, lung, spleen, heart, kidney, brain and eye. The evaluation of the whole section was semi-quantitatively analyzed for the presence of mineral deposition in the fetal membranes (chorioallantois). Ram tissues analyzed included accessory sex glands, testes, kidney, adrenal glands, bladder, liver, spleen, and inguinal lymph nodes.

### 2.5. Virus Isolation from Animal Tissue

Supernatant from individual tissue homogenates was passed through a 0.22 µm syringe filter and inoculated into duplicate T-25 tissue culture flasks of Vero-76 cells grown to 80% confluence; tissue homogenates from individual fetuses were inoculated into duplicate wells of 6-well plates of Vero-76 cells grown to 80% confluence. Inoculum was incubated on the cell cultures for 1 h at 37 °C with 5% CO_2_. Media containing MEM with 1% FBS, penicillin/streptomycin, amphotericin B, and HEPES was added to each flask and cultures were incubated at 37 °C with 5% CO_2_. Female and fetal tissue cultures were observed until D12 after inoculation and checked daily for cytopathic effect (CPE); on D3, D6, D9, and D12 after inoculation, culture supernatant and some cells were removed for RNA extraction. Male tissue cultures were observed until D7 after inoculation and checked daily for CPE; on D3 and D7 after inoculation, half of the monolayer was scraped and removed with the culture supernatant for RNA extraction.

### 2.6. Real-Time PCR

Tissues, PBMC, and cultures were tested for presence of ZIKV nucleic acids by RT-PCR. All tissue and culture samples were extracted using a previously described guanidinium-isothyocyanate-chloroform (TRIzol Reagent, Invitrogen, Carlsbad, CA, USA) protocol [51]. RT-PCR was performed on tissue samples (female: placenta, spleen, liver, inguinal lymph nodes; fetal: brain, umbilicus, spleen/thymus, lung/liver/kidney; male: testes/epididymides, accessory sex glands, spleen, liver, kidney) using the taqman-based CDC diagnostic one-step RT-PCR protocol for ZIKV as previously described [52,53]. RNA from PBMC, female, and fetal cultures was run in duplicate; RNA from all other samples was run in triplicate. If any sample had disagreement between replicates, RT-PCR was repeated on the original sample. Ovine GAPDH was used as an endogenous control for extraction of tissue RNA. For the ram samples, approximate quantification of ZIKV genomic copies per mL of positive sample was accomplished by comparing samples to a standard curve of synthetic ZIKV RNA template (VR-3252SD, ATCC). A known quantity (100 PFU) of pure ZIKV was used as a standardizing control on all plates. 

### 2.7. IgM Capture ELISA

An ovine IgM capture ELISA was developed based on previous methods for ZIKV IgM detection and an IgM capture ELISA protocol developed for WNV infection in the equine, as previously described [25,52,54]. Briefly, plates were coated with 10 µg polyclonal rabbit anti-sheep IgM (Bethyl Laboratories, Inc., Montgomery, TX, USA) in carbonate/bicarbonate buffer (pH 9.6) and incubated overnight at 4 °C. After blocking with PBS with 2% equine serum and 0.5% Tween20 for 30 min at RT, samples were diluted at 1:400 in blocking buffer (PBS with 5% milk and 0.5% Tween20) and added in duplicate wells for 1 h at 37 °C. After washing, ZIKV virus or negative cell culture, grown and formalin inactivated as previously described [55], was added to each well and incubated for 1 h at 37 °C. After washing, a 1:50 dilution of anti-flavivirus antibody (D1-4G2-4-15, ATCC) in blocking buffer was added to each well and incubated for 1 h at 37 °C. Following incubation, a 1:100 dilution of sheep anti-mouse IgG antibody conjugated with horseradish peroxidase (Bethyl Laboratories, Inc.) in blocking buffer was added to each well and incubated for 1 h at RT. After washing, 3,3′,5,5′tretramethylbensidine (TMB, Neogen Life Sciences, Lexington, KY, USA) was added to each well and incubated for 10 min at RT. The reaction was stopped with 2.8% sulfuric acid and plates were read at 450 nm using a microplate reader (Multiskan FC, Fisher Scientific Instruments, Co., Shanghai, China). An immune status ratio (ISR) was calculated for each specimen (Zika Ag OD450 divided by NCA OD450); positive seroconversion was defined as an ISR ≥ 1.5.

### 2.8. Fluorescence-Activated Cell Sorting

Fluorescence-activated cell sorting (FACS) was used to phenotype PBMC isolated from each ewe over the first two weeks PI; the PBMC phenotype of the rams was not analyzed. Quantitated cells included CD4+ T helper cells, CD8+ T effector cells, γδ T cells, CD14+ monocytes, and B cells. Cells were stained in duplicate using mouse anti-sheep CD45 (1.11.32, MCA2220PE, BioRad, Hercules, CA, USA), rabbit anti-sheep CD3 (SP7, ab16669, Abcam, Cambridge, MA, USA), goat anti-rabbit IgG (RRID AB_2539814, P-10994, Thermo Fisher Scientific), mouse anti-sheep CD4 (44.38, MCA2213A647, BioRad), mouse anti-sheep CD8 (38.65, MCA2216F, BioRad), mouse anti-human CD14 (TÜK4, MCA1568A700T, BioRad), mouse anti-sheep γδ T cell (WC1-N2, Washington State University Monoclonal Antibody Center, Pullman, WA, USA), goat anti-mouse IgG1 (1070-19, Southern Biotech, Birmingham, AL, USA), mouse anti-ruminant B cell (BAQ44A, S-BOV2064, Washington State University Monoclonal Antibody Center), and rat anti-mouse IgM (RMM-1, 406529, Biolegend, Waltham, MA, USA) as previously described [25].

All samples were analyzed using a flow cytometer (LSR Fortessa Flow Cytometer, BD Biosciences, Franklin Lakes, NJ, USA) and commercial software (FACS Diva v8.0.1, BD Biosciences). Immune phenotypes were defined as CD4+ T helper cells (CD45+CD3+B-CD4+CD8-CD14-), CD8+ effector T cells (CD45+CD3+B-CD4-CD8+CD14-), γδ T cells (CD45+CD3+B-γδ+), monocytes (CD45+CD3-B-CD14+), and B cells (CD45+CD3-B+). The data for each phenotype were calculated as the average of each duplicate run, presented as the percentage of a specific cell type within the CD45+ gated whole cells.

### 2.9. Statistical Analysis

Statistical analysis was completed using a commercial software program (MedCalc Software bvba v18.2.1, Ostend, Belgium; http://www.medcalc.org; 2018). Significance was determined by a *p* ≤ 0.05 (α = 0.05). Morphometric data were compared between fetuses from control animals and fetuses from infected animals using a one-way analysis of variance (ANOVA) with a Levene’s test for equality of error variances and a Shapiro–Wilk test for normal distribution of residuals for each morphometric ratio. PBMC phenotype post-infection was analyzed for significance using repeated-measures ANOVA. 

## 3. Results

### 3.1. Pregnant Ewe Infections

#### 3.1.1. Ewe Clinical Disease and Gross Pathology

Placental abnormalities were present on gross post-mortem examination in the infected animals. For each animal, the total number of placentomes was counted and placentomes were characterized as either “Type A, B, C or D,”. Type A is classified as a normal placentome with a majority of maternal tissue (known as the “caruncle”) surrounding fetal tissue (known as the “cotyledon”), Types B consists of 30% inversion exposing the upper rim of the cotyledon, C consists of approximately 50% of inversion with flattening of the placentome, and Type D is a complete inversion of the placentome with fetal tissue completely surrounding maternal tissue [56,57]. The number of the placentomes of each ewe was within normal limits, but all infected animals had at least some degree of placentome inversion (“advanced”) morphology (Figure 3; Table 1). 

In two of the infected animals, all placentomes (100%) exhibited Type D morphology, while the placentomes of the other two infected animals had a smaller percentage of advanced placental morphology. Inverted placentomes were not noted in either of the control ewe placentas.

#### 3.1.2. Ewe Histopathology

Regardless of the differences in the gross morphology of Type A through Type D placentomes (Figure 4A,B), all of the placentome samples analyzed from control and ZIKV-infected animals exhibited similar histological patterns (Figure 4C–F). The placentomes were composed of the maternal plates (caruncle) tightly interlaced with the fetal villi (cotyledon) (Figure 4E,F). The maternal plates were lined by a moderately plump single, cuboidal to columnar epithelium with lacy, palely eosinophilic cytoplasm and basilar oriented, round nuclei with coarse chromatin and variably evident nucleoli supported by a variably dense fibrovascular stroma. Multifocally, the maternal epithelium was markedly attenuated with a small, dense nuclei. These areas of maternal epithelium attenuation were often associated with syncytial formation (Figure 4E,F). The syncytia were scattered throughout the maternal–fetal interface, and were more frequent in the deep regions of the placentomes closer to the myometrium (Figure 4E,F). The fetal villi were composed of chorionic epithelium supported fetal mesenchyme. The chorionic epithelium was mainly composed of columnar to cuboidal, segmentally stratified epithelium with large nuclear vesicular chromatin and unapparent nuclei admixed with scattered foamy giant, bi- or trinucleate trophoblasts with centrally located, moderately dense to coarse chromatin. The fetal mesenchyme was composed of low numbers of spindle cells with undefined cell borders supported by a loose fibrovascular stroma. 

Throughout the maternal–fetal interface, randomly distributed, small to moderately cellular aggregates composed of multiple, dense, pyknotic nuclei with a brightly eosinophilic homogenous cytoplasm (apoptotic/necrotic syncytium, presumptively) were observed. On the arcade region of all the placentomes, there were multifocal hematophagous areas ranging from 2 to 5 per section in all animals (Figure 4C,D). In one of the infected animals (I4), there was a focal area of minimal hemorrhage deeper in the placentome (data not shown).

In all animals, the interplacentomal endometrium was lined by a columnar to cuboidal luminal epithelium with a dense stratum compactum composed of spindle cells (e.g., macrophages and fibroblasts) supported by a moderately dense fibrovascular stroma and stratum spongiosum composed of the numerous uterine glands supported by a loose, edematous fibrovascular stroma and overlying the myometrium. In one ZIKV-infected animal (I2), the interplacentomal endometrium had two foci of lymphoid hyperplasia characterized by a center of epithelioid macrophages surrounded by layers of lymphocytes and fewer plasma cells within the transition between the stratum compactum and stratum spongiosum (Figure 5). Focally, there were few multinucleated giant cells (Langerhans type multinucleated macrophages, presumptively) surrounding the lymphoid aggregates. No significant lesions were observed in the other tissues analyzed. 

The most striking lesion was the mineral deposition observed in the chorioallantoic membrane rather than at the maternal–fetal interface. The mineralization was observed in the superficial fetal mesenchyme below the trophectoderm that formed, multifocal to coalescing, large clusters composed of moderately pale to dark basophilic areas of mineralization which stain dark brown to black with Von Kossa histochemical stain (Figure 6A–I). Frequently, there was multifocal perivascular and mural mineral deposition, often extending through the adventitia and the tunica media of the affected vessels, rarely extending into the intima (Figure 6B,I). There was a segmental, thin band of spindle cells just adjacent to the chorionic and allantoic epithelium that had grainy, dark brown granular intracytoplasmic material (early stages of mineral deposition) (Figure 6G,H). Moreover, there was multifocal and segmental mineral accumulation in the chorionic (Figure 6G) and allantoic (Figure 6I) epithelium of the chorioallantoic membrane (highlighted with Von Kossa stain).

The whole sections of placental tissue from all animals were semi-quantitatively analyzed for the presence of mineral deposition in the fetal membranes (chorioallantois). The level of mineralization was classified as mild: presence of mineral deposits in 0–5% of the whole section (Figure 7A), moderate: presence of mineral deposits in 10–30% of the entire section (Figure 7B), and severe: presence of mineral deposits in over 40% of the whole section (Figure 7A) of Von Kossa stained sections. 

In one control animal (C2), there was mild to moderate mineral accumulation in the fetal mesenchyme of the interplacentomal chorioallantoic membrane and over the chorioallantoic layer over the limbus of the placentome (arcade region). Three of the ZIKV-infected animals (I1 & I3–4) had severe mineral deposition in the fetal membranes mainly within the interplacentomal chorioallantoic membrane (Figure 7 and Table 1).

#### 3.1.3. Fetal Development and Pathology

Fetal number and viability were assessed on a weekly basis using transabdominal ultrasonography; all fetuses remained viable during the course of the infection period. At the time of necropsy, grossly misshapen heads were noticeable in most of the fetuses of infected ewes (Figure 8A). Fetal measurements taken during necropsy confirmed that the CC of infected fetuses (mean = 113.13 mm, SD = 7.43) was significantly smaller than that of control fetuses (mean = 125.67 mm, SD = 8.96) (*p* = 0.042). Fetal measurements were used to calculate morphometric ratios to better assess the relative growth of infected fetuses compared with control fetuses. Of note, the cranial circumference to crown–rump length (CC:CRL) and cranial circumference to nose–occipital length (CC:NOL) were significantly smaller in fetuses of infected ewes when compared with that of control fetuses (*p* = 0.006 and *p* = 0.041, respectively) (Figure 8B,C). 

The histopathological analysis of the fetal tissue revealed that in the brain of one fetus (I3F2) from a ZIKV-infected animal, there was an area of meningeal mineralization with minimal inflammatory infiltrate between the hippocampus and the thalamus. There was rare perivascular cuffing with one to two layers of mononuclear (lymphocytes, presumptively) cells surrounding the small-caliber vessels (Figure 7D,E). No significant histological lesions were observed in the other fetal tissues analyzed.

#### 3.1.4. Virus Isolation and Detection 

Cultures of brain were positive for ZIKV in three fetuses from infected ewes (Table 2). Cultures of umbilicus, spleen/thymus, and lung/liver/kidney were positive in two of these fetuses (Table 2). All infected ewes had RT-PCR positive spleens at the time of necropsy (Table 1). ZIKV was cultured from the placenta of one infected pregnant ewe (Table 1). Additionally, fresh tissue from one infected ewe (placenta) and her fetus (brain) were ZIKV positive by RT-PCR (Table 1; Table 2). This same animal had RT-PCR positive PBMCs on D3 PI (Table 1). Neither control ewe nor their respective fetuses had ZIKV-positive tissue or cultures at any point during the study.

#### 3.1.5. Serology

All ewes were negative for ZIKV IgM prior to beginning the study. Following viral challenge, all infected ewes seroconverted to ZIKV by D6 PI (Figure 9A). Neither control animal experienced a detectable rise in ZIKV-specific IgM titers at any point during the study period.

#### 3.1.6. Ewe PBMC Phenotype

The ante-infection and post-infection PBMC phenotype were analyzed for all ewes in the second experiment. No significant changes in the composition or percentages of specific PBMC types within or between groups were noted (Appendix A).

### 3.2. Male Infections

#### 3.2.1. Clinical Disease and Pathology

Daily physical exams and clinical data obtained revealed no evidence of illness or discomfort in the rams. During the male necropsies, no significant gross abnormalities were noted, and histological evaluation of H&E stained tissues revealed no obvious abnormalities in these animals. Histopathology did not reveal significant lesions in the tissues that were analyzed.

#### 3.2.2. Virus Isolation and Detection 

Samples of testes/epididymides, spleen, and liver from rams euthanized on D2, D3, and D6 PI were positive for ZIKV RNA by RT-PCR (Figure 10A). Additionally, ZIKV RNA was also detected from the testes/epididymides, spleen, and accessory sex glands of the ram euthanized on D9 PI. PBMC from the rams euthanized on D6 and D9 were RT-PCR positive between D1 and D5 PI (Figure 10B). Spleen was culture positive in the ram euthanized on D2 PI, while testes from the rams euthanized on D6 and D9 were culture positive (Figure 10A). Neither of the animals euthanized on D4 or D5 PI had positive organs by RT-PCR or culture. Neither control ram had ZIKV-positive tissue or cultures at any point during the study.

#### 3.2.3. Serology

Prior to viral challenge, all rams were negative for ZIKV-specific IgM antibody. The male sheep euthanized on D6 and D9 PI both had positive ZIKV-specific IgM responses beginning by D5 PI (Figure 9B). Prior to D5 PI, both animals had a mean ISR between 1.0 and 1.1. The ISR of both animals continued to rise until their respective days of euthanasia (D6, ISR = 1.47; D9, ISR = 1.75). Neither control animal experienced a detectable rise in ZIKV-specific IgM titers at any point during the study period.

## 4. Discussion

The goals of this study were to determine whether vertical transmission occurred from ewe to fetus following mid-gestation infection, and whether ZIKV could be isolated from the male reproductive tract in ZIKV infected sheep. We have previously illustrated that early gestational infection with ZIKV in pregnant ewes results in maternal infection with significant disruptions to fetal growth patterns [25]. Here, we have expanded upon that model. In the first experiment, infection of pregnant sheep during mid-gestation resulted in viral transmission across the placenta and subsequent fetal malformations. Results from the second experiment indicate that infection of intact male sheep with ZIKV results in productive infection with viral dissemination to the reproductive tract (testes/epididymides, accessory sex glands) as well as highly vascular tissues (spleen, liver). In both experiments, all infected animals tested by ELISA seroconverted with IgM, indicating a ZIKV-specific immune response.

ZIKV is an arbovirus, with the unique ability to be transmitted vertically across the placenta. Furthermore, trimester-specific effects have been noted in humans and NHP models of ZIKV infection during pregnancy, and the timing of infection during pregnancy may influence the severity of fetal effects [2,58,59,60,61]. We have previously shown that early gestation ZIKV infection of sheep results in fetal effects, even in the absence of detectable virus at the tissue level [25]. Here, we have expanded upon that work, showing that mid-gestation ZIKV infection of sheep results in significant changes to fetal development. In the present experiments, fetal morphometric data suggest that growth lagged in the infected fetuses compared with the control fetuses. Although significant, the fetuses were evaluated at 2 weeks PI as opposed to our earlier study, where offspring were evaluated at 6 weeks PI. In the lamb, skeletal growth is extremely rapid during mid-gestation. According to Harris 1937, the basi-sphenoid and occipital bones begin the process of ossification approximately 40 DG and most centers have ossified by 60 DG, with accelerated growth commencing [62]. Between 40 and 80 DG, the CRL expands 5X, from approximately 40 to 200 mm in length. Thus, given that fetuses were evaluated at the time of ossification coupled with rapid growth, significant differences were quantifiable even in this small and short-term study. 

All but one infected fetus (I4F2) had significantly smaller CC compared with the control fetuses. Similarly, ratios of morphometric data, calculated to allow for comparisons in relative growth between fetuses, indicated that the CC:CRL and the CC:NOL were significantly smaller in the infected fetuses compared with control fetuses; thus, the presence of interrupted fetal growth and comparatively smaller head sizes is apparent, similar to our previous findings. In addition, infected fetuses had visible changes in cranial shape and half of these fetuses (I2F1, I2F2, I3F1, I4F1) were positive for ZIKV by culture or RT-PCR with one (I3F2) having mineralization within the leptomeninges. These morphological changes are consistent with those seen in natural human fetal infection as well as experimental ZIKV NHP infections [11,63,64,65]. Mineralization within the brain is a well-described characteristic of CZS in humans [3,4,66,67,68,69,70,71,72], and the mineralization in the leptomeninges of one of the infected fetuses with small cranial circumference strongly suggest that the histopathological changes are due to infection status.

In contrast to our previous work with ZIKV in the sheep model, placental pathology in the present study was much more pronounced. In all infected animals, a large number of inverted, or “advanced,” placentomes was noted, where both control animals had completely normal, un-inverted placentomes. Although the exact etiology of advanced placentomes remains unclear, previous studies have indicated that normal control animals typically have a much higher proportion of Type A placentomes compared with animals that have been placed under stressful metabolic conditions during pregnancy [57,73,74,75]. Thus, it has been hypothesized that the increase in fetal tissue in placentomes with Types B–D morphology may be a compensatory mechanism in cases of placental insufficiency and/or fetal stress resulting from adverse conditions in the ewe [56,58,73]. 

ZIKV associated placental insufficiency has been reported in humans as well as in murine and NHP models of infection [58,76,77,78,79]. It is noteworthy that only infected ewes displayed some level of advanced type placentomes, with the highest numbers of Type C and D placentomes occurring in the animals with fetuses that had the greatest number of culture-positive tissues. All infected ewes also had some degree of histopathology consistent with inflammation or disease processes, and two of these animals (I2 and I4) had ZIKV-positive placentas by culture or RT-PCR. Interestingly, the animal with culture-positive placenta (I2) had 100% Type D placentome morphology with well-organized inflammatory infiltrate within the endometrium while having the greatest number of fetuses with the most ZIKV culture-positive organs. The animal with RT-PCR positive placenta also had 100% Type D placentomes with severe mineralization of the placenta, as well the one fetus with an RT-PCR positive brain. The increased number of advanced placentomes in these ewes could be indicative of virally induced placental insufficiency and future studies are warranted to discern the degree of metabolic disruption at the maternal–fetal interface. In our previous pregnant ewe study, early gestation infection did not result in ZIKV-positive fetuses or advanced placental morphology. 

While some level of necrosis and mineralization of the placenta is considered incidental in ruminants, the severity of placental mineralization in the infected ewes of this study was striking. In the present study, one control animal (C2) did have mild to moderate multifocal mineralization in the fetal mesenchyme; however, it is worth noting that in this animal, no additional abnormal findings were present in either the dam or the fetus. In contrast, all infected ewes in this study exhibited an increased mineralization and/or inflammation of the reproductive tract coupled with the highest detectable virus, most severe pathology, and highest IgM titers. Placental calcification is a common phenomenon in human placenta which increases with gestational age. In humans, degree of placental calcification based on placental maturity as assessed by ultrasonography is graded (Granum scale) and used to identify high risk pregnancies [80]. In addition, increased placental calcification is considered to be evidence of viral infection during human pregnancy and increased calcification has been demonstrated in the placentas of ZIKV infected women [81], as well as in experimentally infected NHPs [15,82]. In ruminants, some placental mineralization may occur in normal animals due to the high cellular turnover in this tissue, but grading schemes for degree of calcification, which predict disease, are not available [83]. Nonetheless, calcification of placenta, as a pathological finding has been associated with BDV gestational infections in the ewe [84]. Although our findings are qualitative, the placental calcification observed in this current study is abundant in the infected ewes for the gestational age of the infected fetuses. Coupled with the gross pathology (inverted placentomes) in the infected animals, this model also serves as an investigatory tool for study of ZIKV effects on placental development.

Beyond its ability to be vertically transmitted, ZIKV is also one of the only known human flaviviruses to be transmitted sexually. Furthermore, male to female transmission is thought to play an important role in ZIKV transmission amongst humans [82,84,85,86,87]. Previously published studies have indicated that ZIKV has many cellular targets within the male reproductive tract [28,29,34,35,84]. Since asymptomatic male infection represents an important risk factor for sexual transmission of ZIKV, feasible and translatable animal models are needed to study the effects of ZIKV infection on male reproductive organs. 

Studies undertaken in our lab indicate that fetal sheep testicular cells are highly permissive to ZIKV infection [25], laying the basis for investigation of sheep as a model of male ZIKV infection. In the present study, data collected from the ram infection study suggest that intact male sheep can be infected with ZIKV, and that replicating virus can be isolated from the testes/epididymides. In this study, ZIKV was isolated from the reproductive tract and other highly vascular tissue (liver, spleen) in two thirds of the infected rams. Interestingly, the largest quantities of ZIKV detected in reproductive tissue by RT-PCR and culture were found in the animals euthanized on D6 and D9 PI. This begs the question of whether increased time is necessary for ZIKV to disseminate to the reproductive tract before replicating to detectable quantities, ultimately leading to persistence of viable ZIKV in male reproductive tissue. 

In NHP models, ZIKV has been shown to localize to the male reproductive tract, infecting the Sertoli cells of the testes, the seminal vesicles, and the prostate [14,48,49,88]; however, some studies fail to show dissemination of virus to the male reproductive tract in macaques. In some NHP studies, virus was found in the urogenital tract, with localization in the kidneys suggesting that ZIKV may travel from the urine to the other organs of the reproductive tract [48,49]. In contrast, ZIKV was not isolated from the kidneys of the rams in our study. This may be due to timing of our examination or may indicate that ZIKV may have preferentially disseminated to the reproductive organs. In NHP models, ZIKV has been shown to persist in the accessory sex glands for over a month post-infection. Future studies are needed to deduce a more concrete timeline of ZIKV dissemination and replication within the male ovine reproductive tract.

In the ram study, ZIKV RNA was not detectable from tissues of the animals euthanized on D4 and D5 PI. Since antibody response and presence of viremia were not measured in these animals, it is not possible to ascertain whether they became infected or whether viral dissemination simply did not occur. In the rams euthanized on D6 and D9 PI, whole blood was collected over the first five days PI to determine whether an immune response and/or viremia could be detected in response to infection; however, blood was not collected from the remaining infected sheep because there was limited time between infection and necropsy (thus, viremia and immune response would likely not be detectable). Identification of ZIKV in circulating PBMC coupled with continuous rise in IgM over the course of the observation period further implicates viremia and successful infection in the rams euthanized on D6 and D9 PI. 

Ultimately, the results of these two experiments coupled with our previous work in the early gestation pregnant ewes indicates that sheep may offer a holistic outbred animal model in which to study pathogenesis of ZIKV in both males and females and may pave the way for future sexual transmission studies.

## Figures and Tables

**Figure 1 viruses-12-00291-f001:**
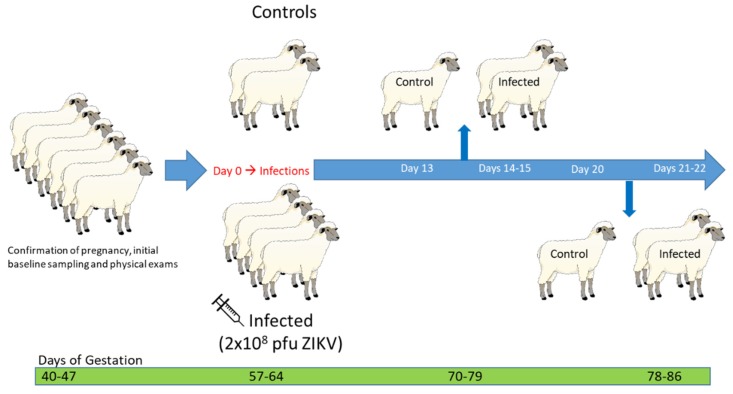
Experimental design of Zika virus (ZIKV) infection of mid-gestation pregnant ewes. Six specific pathogen-free (SPF), mid-gestation pregnant, Polypay ewes were used in this experiment and confirmed pregnant by transabdominal ultrasonography upon receipt. On day D0 post-infection (PI), corresponding with 57–64 DG, four ewes were inoculated with two injections of 1 × 10^8^ PFU ZIKV; two ewes were used as non-infected controls. Whole blood and serum were collected daily from all ewes over the first week PI. Control animals were euthanized on D13 and D20 PI; one infected animal was euthanized on D14, D15, D21, and D22 PI. Necropsies of ewes and fetuses were immediately performed on each animal following euthanasia.

**Figure 2 viruses-12-00291-f002:**
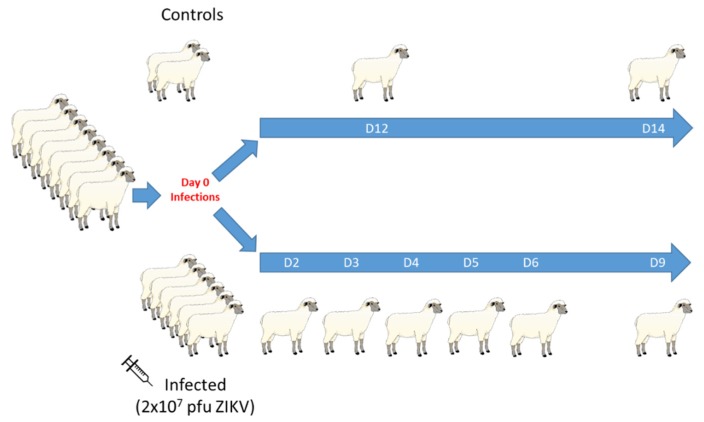
Experimental design of intact ram ZIKV infection study. Eight SPF, sexually mature, intact, Polypay rams were used in this experiment. On day D0 PI, six rams were inoculated with two injections of 1 × 10^7^ PFU ZIKV; two rams were used as non-infected controls. One infected ram was euthanized each day and immediately necropsied on D2, D3, D4, D5, D6, and D9 PI. The control rams were euthanized on D12 and D14 PI. Necropsies were performed on each ram immediately following euthanasia. Daily whole blood and serum were collected from the two control rams as well as the rams euthanized on D6 and D9 PI.

**Figure 3 viruses-12-00291-f003:**
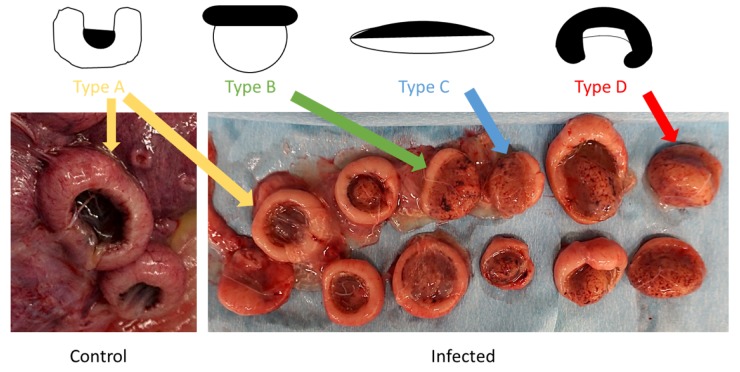
Schematic and photographic images depicting sheep placentome morphology. In the schematic representation of the placentomes, white areas represent maternal tissue, while black areas represent fetal tissue. The schematic images have been adapted from Vatnick et al. (1991) and depict the level of inversion in each placentome type; Type A placentomes have no inversion, Type B placentomes exhibit at least 30% inversion, Type C placentomes exhibit approximately 50% inversion or a flat appearance, and Type D placentomes are completely inverted (100%). The photograph on the left depicts healthy Type A placentomes collected from a control animal at the time of necropsy. The photograph on the right depicts a range of all morphological placentome types collected from an infected ewe at the time of necropsy.

**Figure 4 viruses-12-00291-f004:**
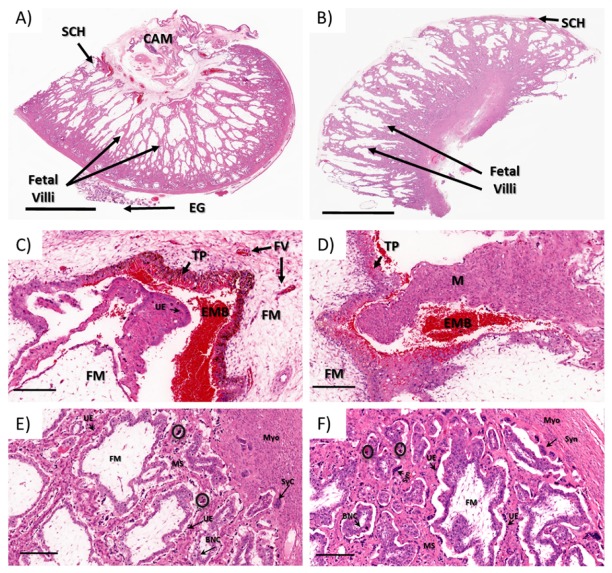
Placentome histology of (**A**,**C**,**E**) control and (**B**,**D**,**F**) ZIKV-infected sheep (H&E). (**A**) Concave placentome (Type A) from a pregnant control ewe; (**B**) inverted placentome (Type D) from a ZIKV-infected ewe; (**C**,**D**) subchorial hematomas in (**C**) control and (**D**) ZIKV-infected ewes. (**E**,**F**) maternal fetal interface in (**E**) control and (**F**) ZIKV-infected ewes. Subchorial hematoma (SCH); chorioallantoic membrane (CA); endometrial glands (EGs); trophoblast with pigment (TP); extravasated maternal blood (EMB); fetal mesenchyme (FM); maternal side (M); fetal villi (FV); myometrium (Myo); syncytium (Syn); uterine epithelium (UE); binucleated giant cells (BNC); maternal stroma (MS); circles: necrotic/apoptotic syncytium. A–B bars: 4 mm; C–F bars: 200 µM.

**Figure 5 viruses-12-00291-f005:**
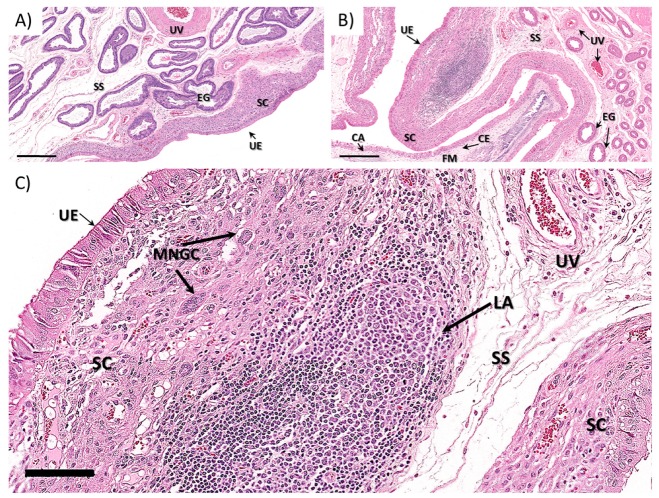
Interplacentomal ovine endometrium from (**A**) control and (**B**,**C**) ZIKV-infected dams (H&E). (**A**) Endometrium with uterine epithelium, stratum compactum, stratum spongiosum and endometrial glands in control ewe. (**B**,**C**) Endometrium with uterine epithelium, stratum compactum, stratum spongiosum and endometrial glands in ZIKV-infected ewe (I2), with well-organized lymphoid aggregate composed of a center of macrophages surrounded by multiple layers of lymphocytes and fewer plasma cells. Additionally, there are fewer giant multinucleated cells (multinucleated, giant macrophages, Langerhan’s type, presumptively) within the stratum compactum close the inflammatory infiltrate. Chorioallantoic membrane (CA); endometrial glands (EGs); fetal mesenchyme (FM); myometrium (Myo); uterine epithelium (UE); stratum compactum (SC); stratum spongiosum (SS); chorionic epithelium (CE) and multinucleate giant cells (MNGCs). Bars: 200 µM.

**Figure 6 viruses-12-00291-f006:**
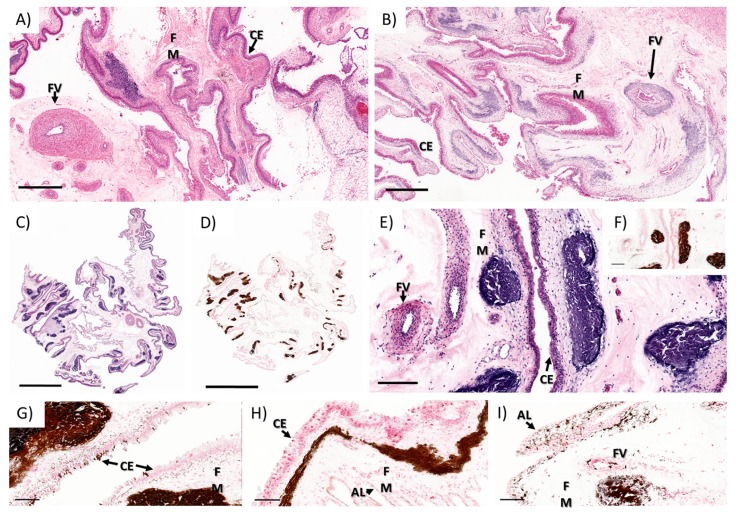
Chorioallantoic membrane from the interplacentomal area from (**A**) control and (**B**) ZIKV-infected ewes (**H**,**E**). (**A**) There are few, multifocal, mild to moderate areas of mineralization on the close to the chorionic epithelium in one control ewe (C2); (**B**) there are multifocal to occasionally coalescing areas of mineralization on the close to the chorionic epithelium and frequently perivascular extending into the adventitia and media of the fetal vessels in ZIKV-infected ewes. (**C**,**E**) At higher magnification, there are multifocal basophilic areas of mineralization (**H**,**E**), which is confirmed in (**D**,**F,G–I**) with Von Kossa special histochemical stain. Chorioallantoic membrane (CA); fetal mesenchyme (FM); chorionic epithelium (CE). **A**,**B** bars: 200 µM; **C**,**D** bars: 4mm; **E**,**F** bars: 50 µM.

**Figure 7 viruses-12-00291-f007:**
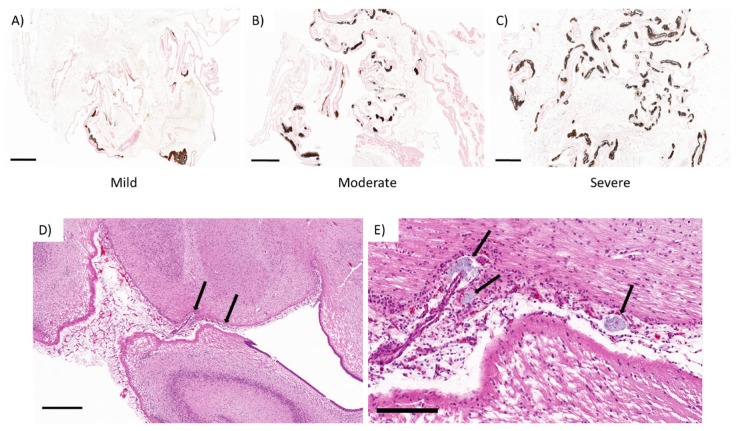
Top: Chorioallantoic membrane from the interplacentomal area, Von Kossa stain. (**A**) Mild, (**B**) moderate, and (**C**) severe mineral accumulation patterns used to evaluate the samples from control and ZIKV-infected ewes. Bottom: Fetal brain; telencephalon between hippocampus and thalamus (I3F2) with multifocal areas of pale basophilic mineral accumulation (black arrows) within the leptomeninges between the hippocampus and the thalamus. (**A**–**D**) bars: 200 µM; (**E**) bar: 50 µM.

**Figure 8 viruses-12-00291-f008:**
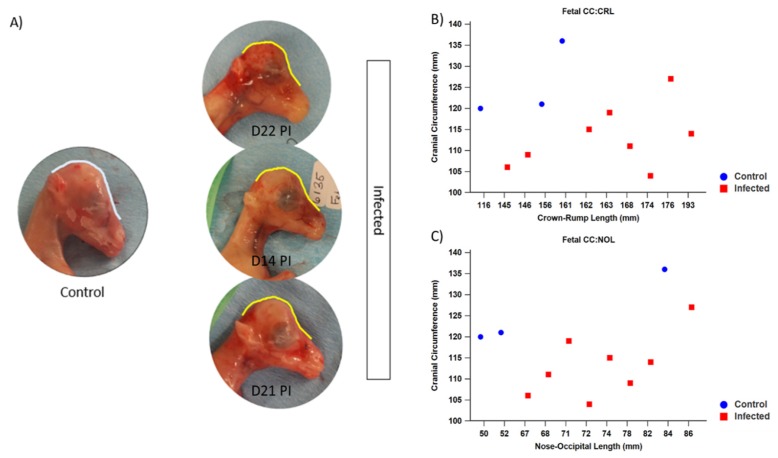
Photographs of fetuses at time of necropsy and fetal measurements taken at the time of necropsy were used to calculate morphometric ratios to better assess relative growth in control and infected fetuses. (**A**) The picture on the left depicts a fetus from a non-infected control ewe; note the outline of the shape of the rostral aspect of the head (light blue) illustrates a smooth contour and no dished nose. The three pictures on the right depict fetuses from infected ewes with obvious changes to the contour of the rostral aspect of the head (outlined in yellow), reminiscent of the phenotypic characteristics seen in human children with microcephaly. (**B**) The cranial circumference (CC) is depicted on the *y*-axis compared with the crown–rump length (CRL) on the *x*-axis; the CC:CRL of infected fetuses was significantly smaller when compared with that of control fetuses (*p* = 0.006). (**C**) The CC is depicted on the *y*-axis compared with the nose–occipital length (NOL); the CC:NOL of infected fetuses was significantly smaller when compared with that of control fetuses (*p* = 0.041).

**Figure 9 viruses-12-00291-f009:**
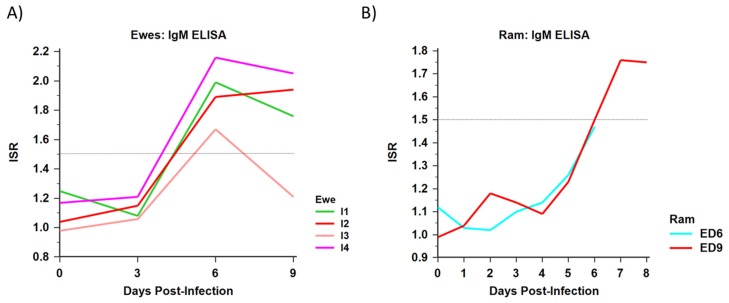
Immune response was determined for infected sheep in both experiments by ZIKV IgM ELISA. (**A**) The immune status ratio (ISR), a representation of IgM antibody response, by days post-infection of all infected ewes. (**B**) The ISR by days post-infection of rams euthanized on D6 and D9 (ED6 and ED9, respectively). In both cases, an ISR ≥ 1.5 is indicative of seroconversion to ZIKV; in both studies, none of the control rams or ewes seroconverted to ZIKV.

**Figure 10 viruses-12-00291-f010:**
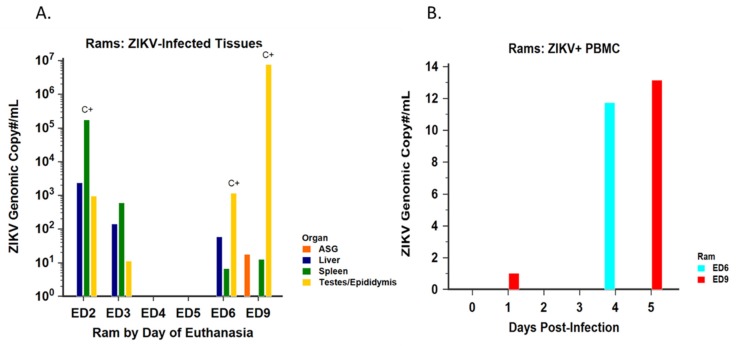
ZIKV-positive tissues in all rams by day of euthanasia (ED). (**A**) Tissues collected at the time of necropsy were cultured and tested for presence of ZIKV RNA; “C+” indicates tissues that were culture positive; bars represent the ZIKV genomic copy number found in RT-PCR+ tissues. (**B**) Blood was collected daily for the first 6 days PI from rams euthanized on D6 and D9; PBMC isolated from whole blood was tested for presence of ZIKV RNA by RT-PCR; bars represent the ZIKV genomic copy number found in PBMC by DPI.

**Table 1 viruses-12-00291-t001:** Summary of abnormal findings at time of necropsy; pregnant ewes.

Ewe ID	Infected?	Fetus(es) with Small Heads	Inverted Placentomes (%)	Placenta	Spleen	PBMC	Mineralization (Placenta)	Histopathological Features
C1	No	0/2	0	-	-	-	None	NSFs ^1^
C2	No	0/1	0	-	-	-	Mild to Moderate	Mineralization present in the FM ^2^ occasionally perivascular closer to the CE ^3^
I1	Yes	2/2	15	-	PCR+	-	Severe	Multifocal areas of mineralization of the superficial FM sometimes around small vessels
I2	Yes	2/2	100	Culture+	PCR+	-	None	Well organized lymphoid aggregates ^4^ surrounded by some MNGCs ^5^ in the IPE ^6^
I3	Yes	2/2	35	-	PCR+	-	Severe	Mineralization present in the FM sometimes around small vessels closer to the CE
I4	Yes	1/2	100	PCR+	PCR+	PCR+	Severe	Mineralization present in the FM sometimes around small vessels closer to the CE, rarely within the PL ^7^

^1^ No significant findings (NSFs), ^2^ Fetal mesenchyme (FM), ^3^ Chorionic epithelium (smooth chorion) (CE), ^4^ Lymphoid aggregates composed of macrophages surrounded by multiple layers of lymphocytes and plasma cells, ^5^ Multinucleated giant cells (MNGCs), ^6^ Interplacentomal endometrium (IPE), ^7^ Placentomal limbus (PL).

**Table 2 viruses-12-00291-t002:** Summary of abnormal findings; fetuses.

Fetus ID	Infection Status	Small Head?	Brain	Brain Mineralization	Spleen/ Thymus	Lung/Liver/ Kidney	Umbilicus
C1F1	-	No	-	-	-	-	-
C1F2	-	No	-	-	-	-	-
C2F1	-	No	-	-	-	-	-
I1F1	+	Yes	-	-	-	-	-
I1F2	+	Yes	-	-	-	-	-
I2F1	+	Yes	Culture+	-	Culture+	Culture+	-
I2F2	+	Yes	Culture+	-	Culture+	-	Culture+
I3F1	+	Yes	Culture+	-	-	-	-
I3F2	+	Yes	-	+	-	-	-
I4F1	+	Yes	PCR+	-	-	-	-
I4F2	+	No	-	-	-	-	-

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
