# Peer review of "Experimental Infection of Mid-Gestation Pregnant Female and Intact Male Sheep with Zika Virus"

_viruses, 2020, doi:10.3390/v12030291_

Round 1

Reviewer 1 Report

In this manuscript Schwarz et al., developed an mid-gestation pregnant female and intact male sheep models with Zika infection. Even though several models were developed for Zika virus including mice, monkey, pig and now Sheep models (in this manuscript) will help us to understand developmental deficits during Zika infection.  Although this great model to put efforts to develop further in parallel with other animal models. One of the major deficiency of this model (from previous published/present manuscript by same group) is not optimized proper virus detection assays by PCR. The reported viral loads were very low and did not conducted properly. Authors should use one of the published taqman based assay including internal housekeeping gene to normalize data. Each of the animals longitudinal plasma viral loads (with copy numbers) needs to be shown first to know whether animals are infected and their sustained viral loads during the infection, this is very important to validate the model as compare other (mice, monkeys etc). Next, each of the tissues needs to carefully evaluated for DNA and RNA loads. The manuscript suffering from proper detection methods.  Even though other serology and IHC data supports that animals are infected but PCR data is very crucial specifically number of copies etc during sexual transmission involved in order to widely accept this model. Therefore, strongly suggested authors to collaborate one of the monkey research groups to run these assays and presented during the revision and make appropriate conclusions.

Most of the times, PMBCs were used to culture and then reported virus positive negative? The direct measurements from tissues using sensitive PCR is warranted.

Minor,

There is no Fig 10, exists and text they referred to Fig 10A? Fig. 1 is cut off,

Some methods were very similar to previously published paper from this group., needs English editing.

Reviewer 2 Report

This is a nice paper with a significant amount of work exploring the utility of sheep as an animal model for ZIKV pathogenesis during pregnancy as well as sexual transmission in males.  The work is generally scientifically sound, with logically designed experiments and appropriate techniques.  The paper is largely qualitative and descriptive but this should not detract from its utility.

My main concern is the very low number of control ewes (N=2).  I understand the ethical/ animal care considerations but it is still a very low number to give us some range as to what the placentas should look like.

Figure 8 B and C - it seems that the number of control fetuses is dramatically smaller than the infected ones.  Can the authors explain why that is even though the number of control ewes is half of that infected ewes?  Why would the control ewes produce fewer fetuses? 

Section 3.1.6 - these data, even though not significant, should be shown (at the very least as an appendix).  It is surprising that there is no difference at all between control and ZIKV infected in immune cell activation.  Can the authors explain why? 

In the introduction, can the authors please provide some additional information on the sheep gestational cycle and its similarity to the human one?

Figure 1 needs a bit of editing as the right hand side is missing

Reviewer 3 Report

Comments:

Description: Erika et al., clearly explained vertical transmission of Zika virus infection in the ovine model in continuation with the previous publication. Histological interpretations of the placenta are highlights of this manuscript from experiment 1.  This information very relevant to the researcher who is developing animal models for Zika virus infection.

Major comments:  No major comments.

Minor comments:

With experiment 2 author could have explored further into the infected cell types in the epididymis and the testis would enhance this manuscript, because the presence of ZIKV viral RNA in those tissues has been explained in other models as well cell types have been defined. This may be good information but does not add any new information to the literature.

As authors claiming mineralization is the primary cause, are there any cell and molecular biological evidence that Zika virus-induced infection causes calcification through necrosis. Also, moderate mineralization was reported from uninfected ewe ID C2 in table 1. So mineralization is possible with or without Zika virus infection.

May be providing salient features of the ovine model, in bullet format, compared to other existing models would benefit the reader easily.

Reasons for necropsy days after infection can be better explained in experiment 1, D14, D15, D21, D22. Because the reviewer feels that within 22nd days of infection, authors could be able to see the changes at the CC-CRL values, which is a bit of surprise. 

Is this ovine model lethal or sub-lethal that can be specified?.

Figures 8B and C can be shown in bar graph format and the calculated p values would enhance the data.

Since FACS data did not yield any significant results section 3.1.6, that part can be removed from the manuscript.

It would be an interesting observation that if the study was extended to full term or alternatively up to end of 2nd trimester (100 to 105 days of pregnancy) would provide a greater impact on the CC-CRL and CC-NOL values.

Tables 1 and 2, confusing with “X” on column 2 and column 3 respectively, better use yes, no, and not applicable in words instead of signs. Moreover, “X” means always yes based on our routine day to day activities on filing forms, etc., but here it says negative or no, means C1 is not infected, little confusing or it can be specified at the legend.

Figure 10 is mislabeled as Figure 11 on the legend.

Figure 8, for three infected fetuses, providing dpi on the same gray circle would enhance the details of infection.

Authors can explain in the materials section, why infection performed in two different routes on the same animals, intravenous and subcutaneous, are there any specific reasons to do that?

Round 2

Reviewer 1 Report

In the second round, Authors made their best efforts to answer all the questions and manuscript quality is improved.

Reviewer 2 Report

No further suggestions.